# HIV serologically indeterminate individuals: Future HIV status and risk factors

**George Mwinnyaa**[1], **Mary K. Grabowski**[2], **Ronald H. Gray**[3], **Maria Wawer**[3], **Larry W. Chang**[2], **Joseph Ssekasanvu**[3], **Joseph Kagaayi**[4], **Godfrey Kigozi**[4], **Sarah Kalibbala**[4], **Ronald M. Galiwango**[4], **Anthony Ndyanabo**[4], **David Serwadda**[5], **Thomas C. Quinn**[1], **Steven J. Reynolds**[1], **Oliver Laeyendecker**[1]\*

1 NIAID, Baltimore, MD, United States of America, 2 Johns Hopkins University School of Medicine, Baltimore, MD, United States of America, 3 Johns Hopkins University Bloomberg School of Public Health, Baltimore, MD, United States of America, 4 Rakai Health Sciences Program, Kalisizo, Uganda, 5 Makerere University, Kampala, Uganda

\* olaeyen1@jhmi.edu

## Abstract

### Background

Indeterminate HIV test results are common, but little is known about the evolution of indeterminate serology and its sociodemographic and behavioral correlates. We assessed future HIV serological outcomes for individuals with indeterminate results and associated factors in Rakai, Uganda.

### Methods

115,944 serological results, defined by two enzyme immunoassay (EIAs), among 39,440 individuals aged 15–49 years in the Rakai Community Cohort Study were assessed. Indeterminate results were defined as contradictory EIAs. Modified Poisson regression models with generalized estimating equations were used to assess prevalence ratios (PRs) of subsequent HIV serological outcomes and factors associated with HIV indeterminate results.

### Results

The prevalence of HIV serologically indeterminate results was 4.9%. Indeterminate results were less likely among women than men (adjPR 0.76, 95% CI 0.71,0.81), in unmarried participants than married participants (adjPR 0.92, 95% CI 0.85,99), and in individuals with primary (adjPR 0.90, 95% CI 0.80,1.02), secondary (adjPR 0.83, 95% CI 0.73,0.96) and post-secondary (adjPR 0.75, 95% CI 0.60,0.94) education, relative to no education. The proportions of persons with indeterminate results progressing to HIV positive, negative or indeterminate results in subsequent visits was 5%, 71% and 24%, respectively.

### Conclusion

HIV serologically indeterminate results were associated with gender and marital status. HIV surveillance programs should develop a protocol for reporting individuals with mixed or

**Data Availability Statement:** The datasets generated and/or analyzed during the current study are not publicly available to preserve the confidentiality of the respondents due to the highly

sensitive data collected. These are restricted by a Data Access Committee constituted by Rakai Health Sciences Program and headed by the Director for Research. All data requests are submitted to Dr Godfrey Kigozi (gkigozi@rhsp.org) who chairs the data access committee. Dr Kigozi shares a data request form which is filled by the person requesting for these data. The form has space where the requester documents why there is a need for these data and how they are going to be manipulated. For this study we requested data for the Rakai Commonity Cohort Study round 1 to round 13. We requested data for the following indicators: Marital status, Gender, Education, Place of resident, Religion, number of sexual partners, occupation, age, Malaria infection status. We also requested for serological data including test kit, test results, final HIV results, date of test, sensitivity and specificity of each test kit. In future when another researcher needs the data for this study, it may also be easy to mention our study in the request.

**Funding:** Supported by the National Institute of Mental Health (R01MH107275), the National Institute of Allergy and Infectious Diseases (R01AI110324, U01AI100031, U01AI075115, R01AI110324, R01AI102939, K01AI125086-01), the National Institute of Child Health and Development (RO1HD070769, R01HD050180), and Division of Intramural Research of the National Institute for Allergy and Infectious Diseases, the World Bank, the Doris Duke Charitable Foundation, the Bill & Melinda Gates Foundation (#08113, 22006.02), and the Johns Hopkins University Center for AIDS Research (P30AI094189). The findings and conclusions in this report are those of the authors and do not represent the official position of the funding agencies. The funders had no role in study design, data collection and analysis, decision to publish, or preparation of the manuscript.

**Competing interests:** The authors have declared that no competing interests exist.

persistently indeterminate HIV results on multiple follow-up visits. Most indeterminate results became HIV-negative over time, but follow-up is still needed to detect positive serologies.

## Introduction

Enzyme-linked immunosorbent assays (ELISA) or enzyme immunoassay (EIA) are widely used for the detection of HIV infection. Even with HIV rapid diagnostic tests, ELISAs are the preferred method for confirmatory tests [1, 2].

However, two ELISA tests with different antigen or test properties can produce discordant results with one test negative and the other positive, usually termed as discrepant [3, 4], discordant [5–7] or indeterminate [8]. The World Health Organization (WHO) recommends that discordant results should be retested by the two assays initially used and if the results remain discordant, the serum should be considered indeterminate [9].

The causes of indeterminate results range from factors relating to the individual being tested (such as co-infections or cross reactivity with other proteins), test kits being used, assay processing and the population under study [10]. Celum et al. reports that among males, tetanus boosters in the previous two years and having sexual contact with prostitutes were independently associated with HIV indeterminate results; and among females, parity and autoantibodies were independently associated with indeterminate results [11]. HIV indeterminate results have also been associated with systematic lupus erythematosus, rheumatoid factor and polyclonal gammopathy, antibodies to DR-HLA, cross reactivity to core proteins of other retroviruses, mycobacterium leprae infection, heat inactivation of serum samples, in vitro hemolysis, elevated bilirubin levels and tetanus vaccination [12]. An HIV indeterminate result may be associated with acute HIV infection [13–15], and WHO recommends that such an outcome among high risk populations or those with a history of high risk behaviors should be considered as a potential case of acute HIV infection [16].

The prevalence of HIV indeterminate results varies between study populations, assays and test kits [8]. A review of studies using different test kits shows that the prevalence of HIV indeterminate results ranges from approximately 6% to 50% [10]. Other studies report serologically indeterminate prevalence below 6% [6, 8, 17]. However, it is challenging to project future outcomes from existing studies of HIV indeterminate results due to small sample sizes [12–14] and short follow-up time (less than a year) [8, 11, 18]. Understanding the correlates of ELISA indeterminate results may help HIV surveillance programs to select the appropriate tests and to develop a protocol on how to handle individuals with ELISA indeterminate results.

This study used longitudinal data to assess the long-term outcomes and the within person correlation of indeterminate results. We also explored factors associated with HIV indeterminate results.

## Methods

Data were derived from the Rakai Community Cohort Study (RCCS), a population-based open cohort in 50 communities in rural Rakai District, Uganda. Following a household census enumeration, all consenting individuals aged 15–49 were interviewed by trained same-sex interviewers. Behavioral (e.g. number of sexual partners, alcohol use, religion), health (e.g. malaria infection), demographic (e.g. age, sex, marital status) and socioeconomic (e.g. education level, occupation) information as well as blood samples for HIV testing were consistently

collected across all surveys [19, 20]. The interval between surveys ranged from 12 months to over 18 months.

The present analysis includes 39,440 individuals who provided 115,944 person-visits from 1994 to 2009. HIV was detected by two parallel ELISA test results (See S1 Fig, Supplementary Digital Content 1, which demonstrates the parallel testing algorithm). All HIV testing was done in the same laboratory. The same pair of test kits were used for the first eight surveys. However, a combination of test kits was used for subsequent surveys (for details on test kits used at each survey, their sensitivity and specificity See S3 Table, Supplementary Digital Content 7, which shows the test kits at each round, the sensitivity and specificity and the number of samples tested by each test kit). HIV indeterminate results were defined as discordant ELISA test results (i.e. one positive and one negative). Other variables included in this study were self-reported

Descriptive analysis estimated the proportions of HIV indeterminate results by RCCS visits and covariates of interest. Time-lagged analysis was used to assess the prevalence of subsequent HIV serological outcomes for participants with prior indeterminate, negative or positive HIV serological results among participants with two or more visits (n = 20,000). The HIV status for individuals with two consecutive indeterminate, negative or positive results was further evaluated as a sensitivity analysis. Modified Poisson regression models using generalized estimation equations with robust variance to account for repeated observations were used to examine associations with serologically indeterminate results (for all participants including those with a single visit, n = 39,440). Lorelogram was used to assess the within person correlation of indeterminate results over multiple study visits [21]. A sensitivity analysis was done to assess factors associated with having two or more indeterminate results limited to participants with two or more HIV indeterminate or negative results (n = 20,000). Decision trees were used to determine the trajectory of individuals with HIV negative, positive or indeterminate results at their first visit by tracking their HIV ELISA results for the subsequent three consecutive visits. Associations with a two-sided p-value ≤ 0.05 were considered statistically significant. STATA 14 was used for analysis [22].

The RCCS was approved by Institutional Review Boards (IRBs) in Uganda (The Research and Ethics Committee of the Uganda Virus Research Institute and the Uganda National Council for Science and Technology), and IRBs at Johns Hopkins University and Western IRB. All subjects ≥18 years of age provided written informed consent and minors gave assent with parental/guardian consent to participate in the study. The research was conducted in accordance with the core principles expressed by the Declaration of Helsinki.

## Results

There were 115,944 observations from 39,440 participants. The majority (22,326 (57%)) of the participants were female and 17,114 (43%) were male. The prevalence of HIV indeterminate results over all study visits was 4.9% (5,680/115,944); 5.6% (2,760/4,9695) for males and 4.4% (2,920/66,249) for females. The prevalence of indeterminate results varied by survey round (See S3 Fig, Supplementary Digital Content 8, which shows the prevalence of indeterminate results by survey round).

Females were 24% less likely to test indeterminate compared to males (adjPR 0.76, 95% CI 0.71,0.81) (Table 1). The prevalence of indeterminate results was 5.1% (3,694/71,813) and 4.5% (1,986/44,131) for married and unmarried individuals, respectively; and unmarried participants were 8% less likely to test indeterminate compared to married participants (adjPR 0.92, 95% CI 0.85,0.99). The prevalence of indeterminate results among participants with no education was 5.3% (438/8,303), 5.0% (3,802/75,778), 4.5% (1,272/28,037) and 4.4% (168/3,826) for

**Table 1. Factors associated with HIV serologically indeterminate results among 39440 (115944 person-visits) RCCS participants in Rakai, Uganda (1994–2009).**

| Factors | Observations (%) | EIA Indeterminate$_i$ prevalence | UnadjPR (95% CI) | AdjPR (95% CI)* |
|---|---|---|---|---|
| **Marital Status** | | | | |
| Married | 71813(62) | 3694/71813 = 5.1% | 1.00 | 1.00 |
| Not married | 44131(38) | 1986/44131 = 4.5% | 0.89(0.84,0.95) | 0.92(0.85,0.99) |
| **Gender** | | | | |
| Male | 49695(43) | 2760/49695 = 5.6% | 1.00 | 1.00 |
| Female | 66249(57) | 2920/66249 = 4.4% | 0.79(0.74,0.84) | 0.76(0.71,0.81) |
| **Education** | | | | |
| No education | 8303(7) | 438/8303 = 5.3% | 1.00 | 1.00 |
| Primary | 75778(65) | 3802/75778 = 5.0% | 0.95(0.84,1.07) | 0.90(0.80,1.02) |
| Secondary | 28037(24) | 1272/28037 = 4.5% | 0.87(0.76,0.99) | 0.83(0.73,0.96) |
| Tertiary | 3826(3) | 168/3826 = 4.4% | 0.80(0.65,99) | 0.75(0.60,0.94) |
| **Resident** | | | | |
| Rural | 70759(61) | 3665/70759 = 5.2% | 1.00 | 1.00 |
| Urban/trading | 45185(39) | 2015/45185 = 4.5% | 0.84(0.79,0.90) | 0.92(0.85,0.99) |
| **No. sex partners** | | | | |
| 0 | 19654(17) | 861/19654 = 4.4% | 1.00 | 1.00 |
| 1 | 74904(65) | 3625/74904 = 4.8% | 1.09(1.01,1.19) | 1.01(0.92,1.11) |
| 2 | 14750(13) | 812/14750 = 5.5% | 1.24(1.12,1.37) | 1.03(0.92,1.15) |
| 3 | 4227(4) | 237/4227 = 5.6% | 1.27(1.10,1.48) | 1.04(0.89,1.22) |
| 4 | 1099(1) | 53/1099 = 4.8% | 1.12(0.85,1.48) | 0.92(0.70,1.21) |
| 5+ | 1310(2) | 92/1310 = 7.0% | 1.54(1.23,1.92) | 1.21(0.97,1.51) |
| **Age** | | | | |
| 15–19 | 21876(19) | 986/21876 = 4.5% | 1.00 | 1.00 |
| 20–24 | 25266(22) | 1242/25266 = 4.9% | 1.09(1.00,1.19) | 1.00(0.91,1.10) |
| 25–29 | 23321(20) | 1162/23321 = 5.0% | 1.10(1.00,1.20) | 0.96(0.87,1.07) |
| 30–34 | 17439(15) | 821/17439 = 4.7% | 1.05(0.95,1.16) | 0.91(0.81,1.02) |
| 35–39 | 12425(11) | 627/12425 = 5.1% | 1.15(1.04,1.29) | 0.99(0.87,1.11) |
| 40–49 | 15617(13) | 842/15617 = 5.4% | 1.21(1.09,1.34) | 1.02(0.91,1.15) |
| **Malaria** | | | | |
| Yes | 3552(3) | 112/3552 = 3.2% | 1.00 | 1.00 |
| No | 112392(98) | 5568/112392 = 5.0% | 1.45(1.21,1.73) | 1.23(1.06,1.50) |

*Model also adjusted for occupation, religion, survey round and region of residence, EIA$_J$ = Enzyme-linked Immunoassay indeterminate.

those with primary, secondary and tertiary education, respectively. Compared to participants with no education, those with primary, secondary and tertiary education were less likely to have indeterminate results (adjPR 0.90, 95% CI 0.80,1.02; adjPR 0.83, 95% CI 0.73,0.96 and adjPR 0.75, 95% CI 0.60,0.94, respectively). People who reported not having malaria were more likely to have indeterminate results compared to people who reported having malaria (adjPR 1.23, 95% CI 1.06,1.50). Similarly, people who resided in urban/trading communities were less likely to have indeterminate results compared to people who resided in rural communities (adjPR 0.92, 95% CI 0.85,0.99). The prevalence of indeterminate results was not associated with number of sexual partners and age (Table 1). When the analysis was limited to the first eight rounds, where the testing was performed with the same pair of test kits, the same inferences held (See S4 Table, Supplemental Digital Content 9).

We performed a similar analysis for factors associated with multiple (two or more) indeterminate results. The overall prevalence of two or more HIV serologically indeterminate results

was 6.5% (5,712/8,7945), with 7% (3,822/38,520) for males and 6% (2,890/49,425) for females (adjPR 0.85, 95% CI 0.73,0.98).

Compared to married participants, those not married were less likely to have two or more indeterminate results (adjPR 0.85, 95% CI 0.73,0.98). Similarly, participants who reported not having malaria were more likely to have two or more indeterminate results (adjPR 1.32, 95% CI 1.07,1.64). The prevalence of having two or more indeterminate results was 20% lower comparing participants who reside in urban/trading areas to participants who reside in rural communities (adjPR 0.80, 95%CI 0.68,0.96). A dose response relationship of repeat indeterminate results was observed among age groups. Compared to younger participants, older individuals are more likely to have two or more indeterminate results. The prevalence of having two or more indeterminate results was not associated with other sociodemographic or behavioral factors (See S1 Table, Supplementary Digital Content 2, which shows factors associated with having two or more indeterminate results).

## Prevalence of indeterminate results by number of visits

Table 2 shows the distribution of indeterminate results by number of visits per participant. Of the 15,896 individuals with one visit, 4% (636/15,896) had indeterminate results. The frequency of having at least one indeterminate result increased with the number of visits per participant. For example, 11% (536/4,725) of individuals with three, 23% (381/1,689) for individuals with six, 31% (264/856) for those with 9 visits. Finally, for those individuals with 11 visits 35% (149/427) had at least one indeterminate result.

## Within person correlation of indeterminate results

Fig 1 is a *Lorelogram* which measures the within person correlation of indeterminate results using log odds ratios [21]. Fig 1 suggests that the within person correlation of indeterminate results is autoregressive with individuals being more likely to test indeterminate closer in time to a prior indeterminate result.

**Table 2. Number of EIA indeterminate results by total number of visits per participant.**

| # (D)[a] | Total number of visits | | | | | | | | | | |
|---|---|---|---|---|---|---|---|---|---|---|---|
| | 1 | 2 | 3 | 4 | 5 | 6 | 7 | 8 | 9 | 10 | 11 |
| 0 | 15260 | 7101 | 4188 | 2360 | 1743 | 1308 | 979 | 750 | 592 | 580 | 278 |
| 1 | 636 | 499 | 440 | 366 | 302 | 288 | 212 | 210 | 173 | 183 | 87 |
| 2 | | 81 | 70 | 63 | 89 | 58 | 61 | 57 | 49 | 33 | 35 |
| 3 | | | 26 | 19 | 27 | 21 | 24 | 22 | 24 | 24 | 14 |
| 4 | | | | 3 | 5 | 12 | 10 | 10 | 6 | 12 | 6 |
| 5 | | | | | 0 | 2 | 8 | 1 | 9 | 7 | 3 |
| 6 | | | | | | 0 | 0 | 3 | 3 | 3 | 2 |
| 7 | | | | | | | 0 | 0 | 0 | 0 | 1 |
| 8 | | | | | | | | 0 | 0 | 0 | 1 |
| 9 | | | | | | | | | 0 | 0 | 0 |
| Total # of obs.[b] | 15896 | 15362 | 14175 | 11244 | 10830 | 10134 | 9058 | 8424 | 7704 | 8420 | 4697 |
| Total # of indiv.[c] | 15896 | 7681 | 4725 | 2811 | 2166 | 1689 | 1294 | 1053 | 856 | 842 | 427 |

[a] Number of indeterminates

[b] total number of observations

[c] total number of individuals.

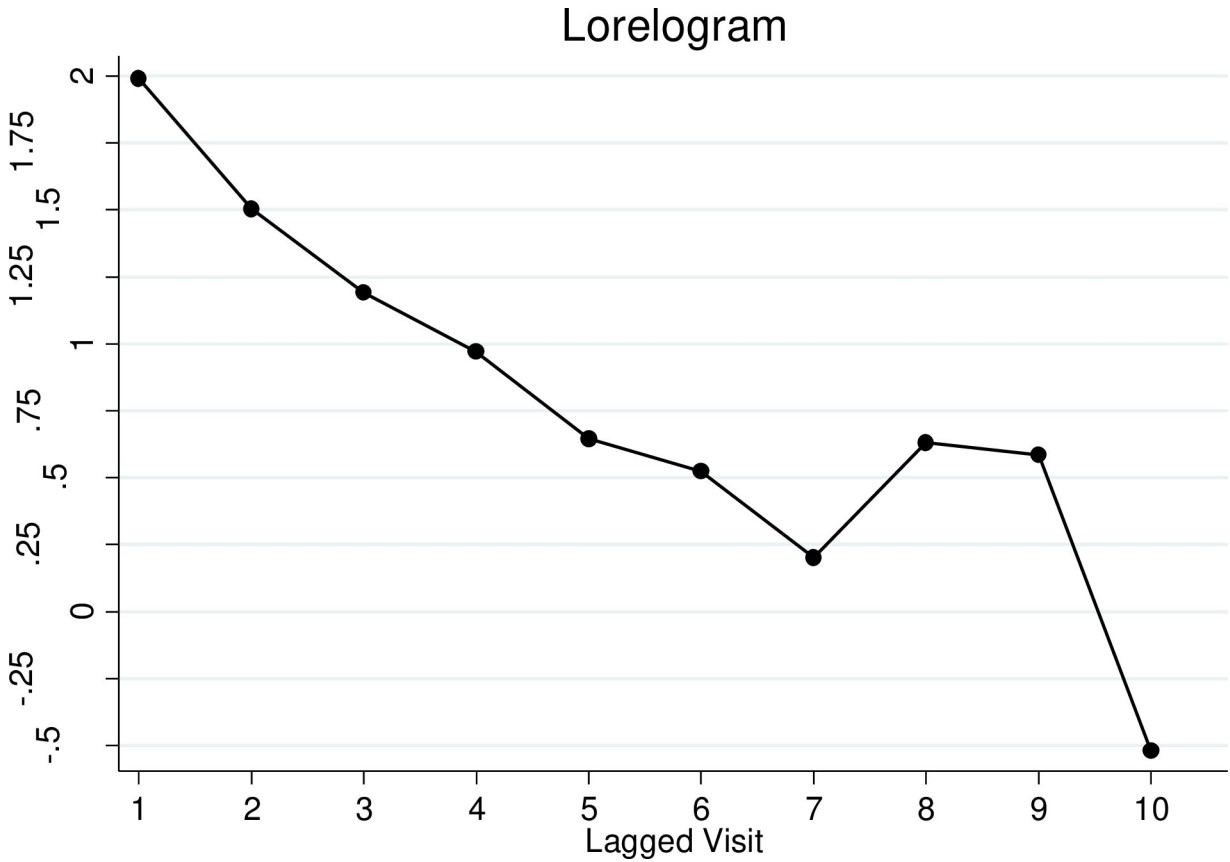

**Fig 1. Lorelogram.** The x-axis is the time-lag between two measurements and the y-axis is log odds ratio.

### Future HIV outcomes for individuals with indeterminate results

We evaluated the probability of transitioning from an indeterminate status to either positive, negative or remaining indeterminate. The proportions transitioning from having an indeterminate result to HIV positive, negative, or remaining indeterminate in subsequent visits were 5%, 71%, and 24%, respectively (Table 3). Participants with indeterminate results who reported having malaria were more likely to transition to HIV positive in subsequent follow-up visits (7/75 = 9%) compared to participants with indeterminate results who reported not having malaria (184/3,852 = 5%). Truck drivers and bar attendants, brewers and hairdressers with indeterminate results were more likely to transition to HIV positive results in subsequent

**Table 3. The prevalence of transitioning from HIV negative, positive or indeterminate to HIV negative, positive or indeterminate in subsequent follow-up serological tests.**

|  |  | HIV+ | HIV- | Indeterminate |
|---|---|---|---|---|
| Lagged result EIA |  |  |  |  |
| Negative | 63743 (83) | 1453/63743 = 2.3% | 5934/63743 = 93.1% | 2949/63743 = 4.6% |
| Indeterminate | 3927 (5) | 191/3927 = 4.9% | 2779/3927 = 70.8% | 957/3927 = 24.4% |
| Positive | 8834 (12) | 8611/8834 = 97.5% | 116/8834 = 1.3% | 107/8834 = 1.2% |

[a] Number of observations, [b] EIA = enzyme immunoassay.

follow-up visits (2/16 = 13% and 4/49 = 8%, respectively) compared to participants in other occupations such as government and salaried employees (11/257 = 4%), students (7/400 = 2%) and agricultural workers (103/2121 = 5%). Similarly, participants with tertiary education who have indeterminate results were less likely to transition to HIV positive in subsequent follow-up visits (6/201 = 3%) compared to participants with primary (128/2,638 = 5%), secondary (41/777 = 5%) or no education (15/310 = 5%). Younger participants (15–19 years) with indeterminate results were less likely to transition to HIV positive in subsequent follow-up visits (8/370 = 2%) compared to participants in older age categories who have indeterminate results (20–24 years 44/813 = 5%; 25–29 years 48/901 = 5%; 30–34 years 35/656 = 5%).

Individuals who tested indeterminate two or more times were more likely to continue to test indeterminate compared to people who had only tested indeterminate once. The percentage transitioning from two consecutive HIV indeterminate results to HIV positive, negative or indeterminate in subsequent visits were 5%, 50% and 46%, respectively. For comparison, for individuals with two consecutive negative results, the proportions transitioning to HIV positive, negative or indeterminate in subsequent visits were 2%, 93% and 5%, respectively. For participants with two consecutive HIV positive results, the proportions transitioning to HIV positive, negative or indeterminate in subsequent visits were 99%, 0.2% and 0.5%, respectively. (See S2 Table, Supplementary Digital Content 3, which shows transitions from two consecutive negative, positive or indeterminate results to HIV positive, negative or indeterminate in subsequent visits).

## HIV serological results trajectory for participants with indeterminate, negative and positive EIA results at their first visit

Of the 39,440 participants in the study 1,667 (4%), 32,867 (83%) and 4,906 (12%) had indeterminate, negative and positive HIV test results at their first visit, respectively (See S2 Fig, Supplementary Digital Content 4, which shows the HIV EIA results at first visit). The HIV results for the subsequent three consecutive follow-up visits for each of these initial serological outcomes at the first visit is illustrated in Fig 2 and Supplementary Digital Content 5 and 6 (S3 and S4 Figs, respectively).

Of the 1,667 participants with indeterminate results at their first visit who had two or more visits (1,031), 22% (229), 72% (744) and 6% (58) had HIV indeterminate, negative and positive

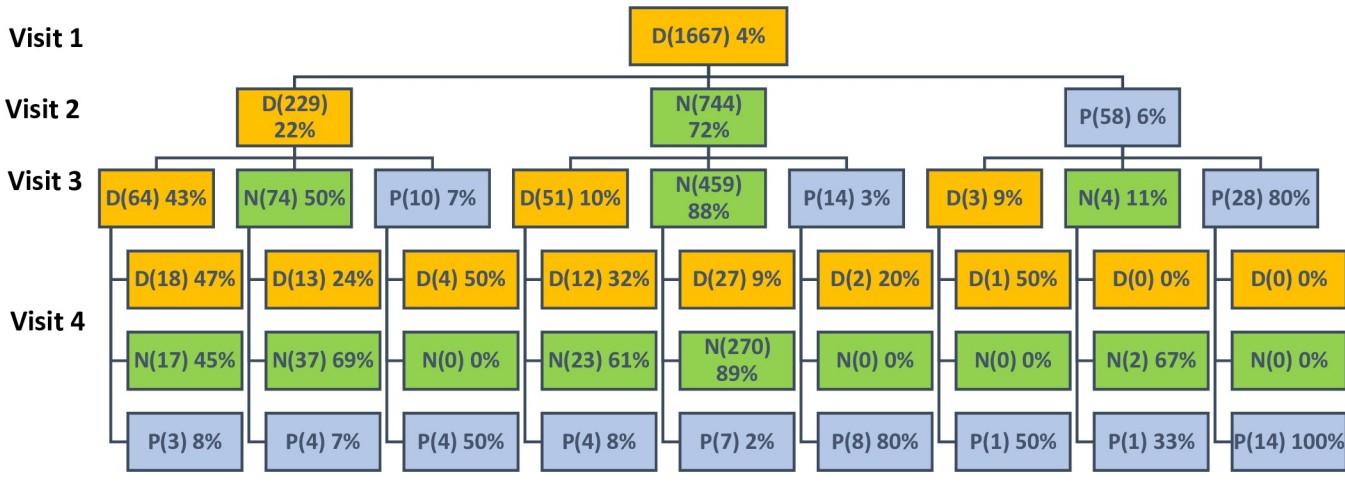

**Fig 2. Future HIV EIA results for individuals with indeterminate results at their first visit.**

results, respectively, at their second visit. Of the 229 participants with indeterminate results at their first and second visit who had three or more visits (148), 43% (64), 50% (74), and 7% (10) had HIV indeterminate, negative and positive results at their third visit, respectively (Fig 2). The trajectory for individuals with positive or negative results at their first visit is illustrated in S3 and S4 Figs (see Supplementary Digital Content 5 and 6, which shows the follow-up serological results for participants with positive or negative HIV results at their first visit). Overall, individuals who tested negative or positive at their first visit were more likely to remain negative or positive at subsequent visits, compared to participants with indeterminate results at their first visit.

## Final HIV status for indeterminate observations

Based on past HIV status and additional confirmatory HIV testing, the final HIV status was determined for each indeterminate observation. Of the 5,680 HIV indeterminate results, 95% (5,411/5,680) were determined to be from HIV negative individuals, and 4% (205/5,680) were determined to be HIV positive individuals. However, the HIV status for the remaining 1% (64/5,680) could not be determined even with additional HIV testing.

## Discussion

We estimate that the overall prevalence of HIV serologically indeterminate based on parallel ELISA testing results in the Rakai Community Cohort Study was 4.9%. We found that indeterminate results were correlated within individuals and that about a quarter of individuals with an initial indeterminate result will have an indeterminate result a year or more later. The proportions having two or more indeterminate results increased with increased age. However, this may reflect a cohort effect: as people get older, they are more likely to participate in multiple surveys. We cannot disentangle the age effect from testing multiple times. The autoregressive within person correlation of indeterminate results, presented in Fig 1, imply that individuals are more likely to test indeterminate closer in time to a prior indeterminate result. To the best of our knowledge, there are no guidelines on how to handle individuals with persistent indeterminate results or mixed results on multiple visits. It is important for specific guidelines to be developed for such individuals. To date, molecular-based tests such as PCR are commonly used to try to resolve discordant HIV ELISA results, which can sometimes give inconclusive results. Also, PCR may not be feasible in remote, hard to reach settings in low- and middle-income countries [23].

The frequency of indeterminate ELISA results was relatively high compared to other studies [4, 7] but comparable to Western Blot indeterminate prevalence [10]. The frequency of transitioning from indeterminate to negative in subsequent visits agrees with findings from other studies that suggest that individuals with indeterminate results should be considered HIV negative if they are not within a high risk group [8, 11, 24–26]. However, the prevalence of transitioning from indeterminate to HIV positive in subsequent visits is not negligible (~5%), which suggests that individuals with indeterminate results may be in the early phase of infection, as suggested by other studies [13–15]. WHO advises that individuals with indeterminate results, particularly in high incidence settings, may be in the acute phase of infection and should receive follow-up testing [16]. A study by Boeras et al. in Rwanda and Zambia reported that 5% of individuals with indeterminate results seroconverted, similar to the result of this study [3].

The findings from this study suggest that individuals with indeterminate results can remain indeterminate for multiple follow-up visits and their HIV status may not be resolved even with years of follow-up. A study by Meles et al. reports that ~94% of individuals who were initially

indeterminate later tested negative during follow-up visits; however, ~7% remained indeterminate after repeated follow up tests. In this study, among participants with three consecutive indeterminate results, ~47%, had another indeterminate EIA result at their fourth visit. This suggests that individuals with persistent indeterminate results are more likely to remain indeterminate. Some of the factors (systematic lupus erythematosus, rheumatoid factor and polyclonal gammopathy, antibodies to DR-HLA, cross reactivity to core proteins of other retroviruses, mycobacterium leprae infection, in vitro hemolysis, parity and tetanus vaccination [12]) known to be associated with indeterminate results are chronic or permanent which in part may explain why certain people may remain indeterminate for a long period of time.

Females, unmarried participants, and individuals with at least primary education were less likely to have indeterminate serologic results. Other studies have demonstrated similar factors correlated with HIV indeterminate results. Carneiro-Proietti et al. reported that 73% of individuals with indeterminate results were male and 54% of all HIV indeterminate individuals were married. Married women in Rakai are less likely to use contraceptives and more likely to have children [27, 28] and parity has been found to be associated with having indeterminate results among females [11]. False HIV EIA positive results has also been reported to be associated with multiparous women [29]. Other factors previously found to be associated with indeterminate serologic results include tetanus immunization in males [11]. However, the prevalence of tetanus immunization among men in Rakai is very low (~23%) [30]. In Uganda, males have higher prevalence of smoking compared to females [31] and it is well established that cigarette smoking can alter blood viscosity [32]. Viscous blood samples or samples with precipitates can form residues which could interfere with ELISA assays [6]. This, in part, may explain why males have higher prevalence of indeterminate results compared to females and it is important for public health screening programs to consider this when screening males for HIV infection.

There are limitations to this study. Test kits that used whole viral lysates are known to have high false positive results and we were not able to stratify our analysis by the antigens used by each test kit [33, 34]. The results presented in this analysis may not hold for other populations, especially those with lower HIV prevalence. In lower prevalence populations, samples with indeterminate results would have a much lower frequency of being true HIV positive. Self-reported malaria infection may not be accurate, and we did not test for other health conditions that have been reported to cause indeterminate results. Additionally, the results from this study should be interpreted with caution given the changes in the HIV epidemic and testing tools over the years. The large sample could also result in statistical significance. Notwithstanding these limitations, this study has significant strength, such as the use of a longitudinal data with parallel ELISA tests done consistently for over 10 years.

The findings from this study are applicable whenever ELISAs are used either for HIV surveillance or clinical diagnoses purposes. In conclusion, females, unmarried individuals and having at least primary education are associated with lower indeterminate results. Individuals with indeterminate results are likely to be HIV negative, but an important proportion (~5%) are found to be HIV positive in subsequent visits. The findings from this study support the WHO recommendation that HIV surveillance programs should analyze and report indeterminate results separately to avoid the over or underestimation of HIV prevalence [9].

## Supporting information

**S1 Fig. Supplemental Digital Content 1: HIV EIA parallel testing algorithm.**
(DOCX)

**S2 Fig. Supplemental Digital Content 4: HIV EIA results at first visit.**
(DOCX)

**S3 Fig. Supplemental Digital Content 5: Future HIV EIA results for individuals with HIV negative results at their first visit.**
(PDF)

**S4 Fig. Supplemental Digital Content 6: Future HIV EIA results for individuals with HIV positive results at their first visit.**
(PDF)

**S5 Fig. Supplemental Digital Content 8: Prevalence of HIV EIA indeterminate results for RCCS participants in Rakai, Uganda (1994–2009).**
(PDF)

**S1 Table. Supplemental Digital Content 2: Factors associated with having two or more HIV serologically indeterminate results among 20000 (87945 person-visits) RCCS participants in Rakai, Uganda (1994–2009).**
(DOCX)

**S2 Table. Supplemental Digital Content 3: The prevalence of transitioning from HIV (NN), (PP) or (DD) to HIV negative (N), positive (P) or indeterminate (D) in subsequent follow-up serological tests.**
(DOCX)

**S3 Table. Supplemental Digital Content 7: Test kit used for each survey round and the percentage of samples tested by each test kit.**
(DOCX)

**S4 Table. Supplemental Digital Content 9: Factors associated with HIV serologically indeterminate results among 26375 (62148 person-visits) RCCS participants in Rakai, Uganda (1994–2002).**
(DOCX)

## Acknowledgments

We thank the staff of the Rakai Health Sciences Program and study participants for their dedication and support.

This study was presented at the Conference on Retroviruses and Opportunistic Infections (CROI), Seattle, March 4–7, 2019.

## Author Contributions

**Conceptualization:** Oliver Laeyendecker.

**Data curation:** Joseph Ssekasanvu, Joseph Kagaayi, Oliver Laeyendecker.

**Formal analysis:** George Mwinnyaa.

**Investigation:** Oliver Laeyendecker.

**Methodology:** Oliver Laeyendecker.

**Resources:** Oliver Laeyendecker.

**Supervision:** Oliver Laeyendecker.

**Writing – original draft:** George Mwinnyaa.

**Writing – review & editing:** George Mwinnyaa, Mary K. Grabowski, Ronald H. Gray, Maria Wawer, Larry W. Chang, Joseph Ssekasanvu, Joseph Kagaayi, Godfrey Kigozi, Sarah Kalibbala, Ronald M. Galiwango, Anthony Ndyanabo, David Serwadda, Thomas C. Quinn, Steven J. Reynolds, Oliver Laeyendecker.

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
