## [Decision Letter · Decision Letter 0]

3 Oct 2019

PONE-D-19-22579

HIV serologically indeterminate individuals: future HIV status and risk factors

PLOS ONE

Dear Dr. Laeyendecker,

Thank you for submitting your manuscript to PLOS ONE. After careful consideration, we feel that it has merit but does not fully meet PLOS ONE’s publication criteria as it currently stands. Therefore, we invite you to submit a revised version of the manuscript that addresses the points raised during the review process.

We would appreciate receiving your revised manuscript by Nov 17 2019 11:59PM. To enhance the reproducibility of your results, we recommend that if applicable you deposit your laboratory protocols in protocols.io, where a protocol can be assigned its own identifier (DOI) such that it can be cited independently in the future. For instructions see: http://journals.plos.org/plosone/s/submission-guidelines#loc-laboratory-protocols

We look forward to receiving your revised manuscript.

Kind regards,

Paul Sandstrom, Ph.D

Academic Editor

PLOS ONE

Journal Requirements:

1. Please provide additional details regarding participant consent. In the ethics statement in the Methods and online submission information, as your study included minors, state whether you obtained consent from parents or guardians. If the need for consent was waived by the ethics committee, please include this information.

Reviewers' comments:

Reviewer's Responses to Questions

**Comments to the Author**

1. Is the manuscript technically sound, and do the data support the conclusions?

Reviewer #1: Yes

2. Has the statistical analysis been performed appropriately and rigorously? 

Reviewer #1: Yes

3. Have the authors made all data underlying the findings in their manuscript fully available?

Reviewer #1: Yes

4. Is the manuscript presented in an intelligible fashion and written in standard English?

Reviewer #1: Yes

5. Review Comments to the Author

Reviewer #1: Indeterminate test results remain a bane of many within the HIV diagnostics testing arena. While this phenomenon is often accepted as a consequence of testing biological specimens, this manuscript has attempted to quantify the impact of this phenomenon using a statistical lens. While very interesting, this reviewer has concerns re: methodology used from the perspective of the test kits used:

1. The number (6) and nature of the tests used in the 13 phases is a bit concerning as far as generalizability is concerned. They are not consistent and while sensitivity remains 100% the specificity does not. The Welcozyme EIA used in the later phases has 99 % specificity. Considering the range of specificities reported by the different manufacturers, this parameter would be expected to have a potentially significant impact on indeterminate (false-reactive) test results.

2. Because of this can the authors stratify their analysis so that the same EIAs used in the parallel testing algorithm are consistent between phases. By admission the authors state that the same pair of EIAs were used in the first 8 phases. Emphasis should be placed on this phase of the test results to avoid generalizing with later phases that employed different EIAs.

3. The stratification should also distinguish the antigens used between the different EIAs. As an example any EIA employing whole-viral lysate will likely increase false-reactives while those employing synthetic peptides/recombinant antigen would be expected to reduce false-reactives. The discussion should also include this variation between the different EIAs as a limitation to their analysis.

4. Regarding generalizability the positive predictive value (PPV) of a screening test/algorithm will be higher in a high-prevalence population. It would be helpful if the authors could consider a sensitivity analysis on what the impact would be in lower-prevalence population such as North America.

5. Lines 265-269. While the authors make the statement that guidelines are lacking on persistently indeterminate (serology) test results, they should make reference to the fact that molecular-based tests remain a commonly used testing strategy to help resolve these problematic specimens.

6. Line 256. Spelling/grammar; determine(d), the authors should do a better spellcheck as the manuscript is peppered with spelling errors.

7. Line 147. There is an interesting finding re: lower indeterminate result in those residing in urban/trading vs rural residence. Could the authors speculate on why this may be ?

6. PLOS authors have the option to publish the peer review history of their article (what does this mean?). If published, this will include your full peer review and any attached files.

Reviewer #1: No

---

## [Author Response · Author response to Decision Letter 0]

16 Jan 2020

The Editors, PLOS ONE

Editorial Office

November 26, 2019

Dear Editor,

On behalf of my co-authors, I resubmit the enclosed manuscript under the new title, “HIV serologically indeterminate individuals: future HIV status and risk factors” to be considered for publication in PLOS ONE. We thank the editors and reviewers for their valuable comments and suggestions. We provide an edited and clean version of the manuscript. We believe the present manuscript is greatly improved and we thank you for considering our work for publication.

Below is a point-by-point address of our responses to the reviewers.

Sincerely,

Oliver Laeyendecker

Oliver Laeyendecker MS, MBA, PhD

Staff Scientist, NIAID, NIH

Assistant Professor of Medicine, SOM, JHU

Assistant Professor of Epidemiology, JHSPH

855 North Wolfe St. 

Rangos Building, room 538A

Baltimore MD, 21205

Phone: 410-502-3268

Email: olaeyen1@jhmi.edu

 

Editor’s comment . Please provide additional details regarding participant consent. In the ethics statement in the Methods and online submission information, as your study included minors, state whether you obtained consent from parents or guardians. If the need for consent was waived by the ethics committee, please include this information.

Response: We thank you for drawing our attention to this important information. The ethics statement has been revised as follows “All subjects ≥18 years of age provided written informed consent and minors gave assent with parental/guardian consent to participate in the study.” 

Review Comments to the Author

Reviewer #1: Indeterminate test results remain a bane of many within the HIV diagnostics testing arena. While this phenomenon is often accepted as a consequence of testing biological specimens, this manuscript has attempted to quantify the impact of this phenomenon using a statistical lens. While very interesting, this reviewer has concerns re: methodology used from the perspective of the test kits used:

The number (6) and nature of the tests used in the 13 phases is a bit concerning as far as generalizability is concerned. They are not consistent and while sensitivity remains 100% the specificity does not. The Welcozyme EIA used in the later phases has 99 % specificity. Considering the range of specificities reported by the different manufacturers, this parameter would be expected to have a potentially significant impact on indeterminate (false-reactive) test results.

Response: We thank the reviewer for their comment and are addressing these issues below. 

2. Because of this can the authors stratify their analysis so that the same EIAs used in the parallel testing algorithm are consistent between phases. By admission the authors state that the same pair of EIAs were used in the first 8 phases. Emphasis should be placed on this phase of the test results to avoid generalizing with later phases that employed different EIAs.

Response: To determine how the change in test kit over time might impact our results, we did a sensitivity analysis limiting the study to round 1 through round 8 where the same pair of test kits were used. The inference did not change when we limited the analysis to these rounds (See supplementary Table 4., Supplemental Digital Content 9). We added the following sentence to the end of the second paragraph of the results section, “When the analysis was limited to the first eight rounds, where the testing was performed with the same pair of kits, the same inferences held (See supplementary Table 4., Supplemental Digital Content 9).”

3. The stratification should also distinguish the antigens used between the different EIAs. As an example any EIA employing whole-viral lysate will likely increase false-reactives while those employing synthetic peptides/recombinant antigen would be expected to reduce false-reactives. The discussion should also include this variation between the different EIAs as a limitation to their analysis.

Response: We thank the reviewer for drawing our attention to this important point. We have added the antigens for each test kits into the supplemental digital content 7, which now shows the test kit, the manufacturers reported sensitivity and specificity and the antigens as well. Due to the nature of the testing we are not able to stratify our analysis by test kit antigens. This will be taken into consideration in future studies when it is possible to stratify by antigens. 

In addition, we have added the following sentence to the limitations “Test kits that used whole viral lysates are known to have high false positive rate and we were not able to stratify our analysis by the antigens used by each test kit.” 

4. Regarding generalizability the positive predictive value (PPV) of a screening test/algorithm will be higher in a high-prevalence population. It would be helpful if the authors could consider a sensitivity analysis on what the impact would be in lower-prevalence population such as North America.

Response: Unfortunately, we do not have a low prevalence population to directly compare our results to. That said, as the proportion of false positives would increase for each test in a lower prevalence population, the proportion of indeterminate results from truly in infected individuals would increase. We have added the following sentence to the limitations paragraph in the discussion. “The results presented in this analysis may not hold for other populations, especially those with lower HIV prevalence. In lower prevalence populations, samples with indeterminate results would have a much lower frequency of being true HIV positive.”

5. Lines 265-269. While the authors make the statement that guidelines are lacking on persistently indeterminate (serology) test results, they should make reference to the fact that molecular-based tests remain a commonly used testing strategy to help resolve these problematic specimens.

Response: We thank the reviewer for the suggestion. We have added a sentence to the first paragraph of the discussion that states: “To date, molecular-based test such as PCR are commonly used to try to resolve discordant HIV ELISA results, which sometimes can still give inconclusive results. Also, PCR may not be feasible in remote hard to reach settings in low and middle income countries.”

6. Line 256. Spelling/grammar; determine(d), the authors should do a better spellcheck as the manuscript is peppered with spelling errors.

Response: We thank the reviewer for their careful reading of the paper and have made corrections to the document.

7. Line 147. There is an interesting finding re: lower indeterminate result in those residing in urban/trading vs rural residence. Could the authors speculate on why this may be?

Response: We speculate that people who reside in rural communities in Uganda are mostly farmers who are exposed to tetanus (from farm related injuries) and other infections as a result of their farming activities. Such individuals’ immune systems would be more exposed to a broad array of infections and likely generate more cross-reactive antibodies. This in part may explain why we see high prevalence of indeterminates among people in rural areas compared to people who reside in urban/trading areas.

---

## [Decision Letter · Decision Letter 1]

8 Jun 2020

PONE-D-19-22579R1

HIV serologically indeterminate individuals: future HIV status and risk factors

PLOS ONE

Dear Dr. Laeyendecker,

Thank you for submitting your manuscript to PLOS ONE. After careful consideration, we feel that it has merit but does not fully meet PLOS ONE’s publication criteria as it currently stands. Therefore, we invite you to submit a revised version of the manuscript that addresses the points raised during the review process.

We look forward to receiving your revised manuscript.

Kind regards,

Zixin Wang, PhD.

Academic Editor

PLOS ONE

Reviewers' comments:

Reviewer's Responses to Questions

**Comments to the Author**

1. If the authors have adequately addressed your comments raised in a previous round of review and you feel that this manuscript is now acceptable for publication, you may indicate that here to bypass the “Comments to the Author” section, enter your conflict of interest statement in the “Confidential to Editor” section, and submit your "Accept" recommendation.

Reviewer #1: All comments have been addressed

Reviewer #2: (No Response)

2. Is the manuscript technically sound, and do the data support the conclusions?

Reviewer #1: (No Response)

Reviewer #2: Yes

3. Has the statistical analysis been performed appropriately and rigorously? 

Reviewer #1: (No Response)

Reviewer #2: Yes

4. Have the authors made all data underlying the findings in their manuscript fully available?

Reviewer #1: (No Response)

Reviewer #2: Yes

5. Is the manuscript presented in an intelligible fashion and written in standard English?

Reviewer #1: (No Response)

Reviewer #2: Yes

6. Review Comments to the Author

Reviewer #1: (No Response)

Reviewer #2: This paper is quite interesting and presents an interesting phenomenon worthy of publication and reflection. Some significant gaps, however, likely first need address.

Major comments

1. This issue of indeterminate results seems focused on EIA and ELISAs; however, both of those test types are very rarely used for clinical management of patients. Instead RDTs are used. It would be helpful to understand and discuss if this might be a phenomenon for RDTs in their current contexts. Further, it wasn’t entirely clear whether the focus and suggested consideration of these results should be primarily or solely in surveillance studies or actual clinical management and diagnosis.

2. It would be very helpful to indicate somewhere (partly in the abstract, but definitely in the methods) the variety of different behavioral, health, demographic and socioeconomic information collected from each patient. Also, whether this was consistently collected across time points.

3. Some of the more important references provided throughout the introduction are incredibly old (+10 years) and should be updated or else the implications lightened. For example, the WHO recommendation is from 1997 and unlikely to still exist and be relevant. This study itself was done between 1994 and 2009, ending over 10 years ago. This absolutely must be included as a limitation and perhaps clearly and specifically discussed with reference to the results – are/can they still be considered valid given the significant changes in policy and the epidemic since then. Further, it’s unclear if the patients included were known HIV-infected and whether on treatment. If not, that changes the implications of the results given the population of HIV-uninfected and untreated is getting increasingly smaller.

4. In the introduction, a number of similar previous works are referenced; however, neither there nor more importantly in the discussion are these discussed and the importance of the current work to that body touched on.

5. There were several interesting findings, many of which were identified previously, such as males and those married were at higher risk of indeterminate rates. However, this should be discussed further. Why might these issues be happening in those populations?

6. It would also be helpful to understand if any combinations of the factors may result in more indeterminate results. For example, older married males or need those factors always be disaggregated?

7. I’m not sure lines 219-238 bring a lot to the results given all others. I might suggest removing, particularly to include a possible analysis per point #6.

8. It would also be helpful to more clearly discuss possible reasons for why indeterminate results may be persistent. What could be causing this? Lines 272-279. Also, what can or should be done for those patients?

9. It also might be worth reflecting on the large sample size. No sample size calculation was included and though some of the differences are relatively small, the incredibly large sample size would cause significance.

Minor comments

1. Line 36: ‘serological’ results rather than ‘serologic’. Also, ‘immunoassays’ is likely the best terminology.

2. The suggestion that those with primary education are less likely to have indeterminate results should be more carefully considered given the adjPR crosses 1.

3. Line 58, ‘assay’ should likely be ‘assays’ for both.

4. Lines 189-206 should be double-spaced.

5. Line 259-260: I’m not sure that’s still the case given significant scale-up of HIV viral load testing and tuberculosis testing, both molecular PCR-based.

7. PLOS authors have the option to publish the peer review history of their article (what does this mean?). If published, this will include your full peer review and any attached files.

Reviewer #1: No

Reviewer #2: No

---

## [Author Response · Author response to Decision Letter 1]

16 Jul 2020

Dear Editor,

Thank you for the opportunity to revise and resubmit our manuscript, HIV serologically indeterminate individuals: future HIV status and risk factors,” for consideration for publication in the PLOS ONE journal. Please find below a point-by-point response to the reviewers’ comments and corresponding changes made to the manuscript. We have revised the manuscript incorporating nearly all of the reviewer’s recommendations and agree that these revisions have strengthened the quality of the article. Please let us know if we can be of any additional assistance in the review of our manuscript. 

Thank you again for your consideration.

Reviewer #1: (No Response)

Reviewer #2: This paper is quite interesting and presents an interesting phenomenon worthy of publication and reflection. Some significant gaps, however, likely first need address.

Major comments

1. This issue of indeterminate results seems focused on EIA and ELISAs; however, both of those test types are very rarely used for clinical management of patients. Instead RDTs are used. It would be helpful to understand and discuss if this might be a phenomenon for RDTs in their current contexts. Further, it wasn’t entirely clear whether the focus and suggested consideration of these results should be primarily or solely in surveillance studies or actual clinical management and diagnosis.

Response: We thank the reviewer for their thoughtful comment. The reviewer is right, the paper is focused on EIA and ELISAs and not on RDTs. Since the paper did not look at RDTs it is impossible to determine if our findings will be applicable to RDTs. ELISAs are used both for surveillance and clinical diagnoses hence we cannot limit our findings to only clinical or surveillance scenarios. In any situation (surveillance or clinical) where EIA and ELISAs are used in the process, our findings may be applicable. In future we plan to examine indeterminate results for RDTs.

We added the following sentence to the manuscript to address this comment “The findings from this study are applicable whenever ELISAs are used either for HIV surveillance or clinical diagnoses purposes.”

2. It would be very helpful to indicate somewhere (partly in the abstract, but definitely in the methods) the variety of different behavioral, health, demographic and socioeconomic information collected from each patient. Also, whether this was consistently collected across time points.

Response: We thank the reviewer for drawing our attention to this important point. The following sentence has been modified to capture both the information collected and the idea that this was consistent over all surveys. We provided reference to earlier publications from RCCs which include details on all the behavioral, health, demographic and socioeconomic information collected from each participant. 

“Behavioral (e.g. number of sexual partners, alcohol use, religion), health (e.g. malaria infection), demographic (e.g. age, sex, marital status) and socioeconomic (e.g. education level, occupation) information as well as blood samples for HIV testing were consistently collected across all surveys(1,2).”

3. Some of the more important references provided throughout the introduction are incredibly old (+10 years) and should be updated or else the implications lightened. For example, the WHO recommendation is from 1997 and unlikely to still exist and be relevant. This study itself was done between 1994 and 2009, ending over 10 years ago. This absolutely must be included as a limitation and perhaps clearly and specifically discussed with reference to the results – are/can they still be considered valid given the significant changes in policy and the epidemic since then. Further, it’s unclear if the patients included were known HIV-infected and whether on treatment. If not, that changes the implications of the results given the population of HIV-uninfected and untreated is getting increasingly smaller. 

Response: We agree with the reviewer that some of the citations are old and that the data for the study itself is old. However, current information on parallel HIV ELISAs are rare, yet these assays are still used for HIV testing today and the issue of indeterminate results persist. Updated information on HIV ELISA indeterminate results are quite limited in the literature which motivated us to carry out our analysis. We indicated the time period (as correctly reference by the reviewer) when the data for our analysis was collected, the test kits and other details to give readers the opportunity to decide whether our findings will be applicable in their situation. 

Treatment became available in Rakai Uganda as part of PEPFAR in 2006. For the time period of this study individuals who had a CD4 count <250 cells/mm3 would be eligible for treatment. When we limited the analysis to the first 8 rounds which occurred between 1994 to 2002 when ART was not available in Rakai, the inference did not change. (See supplementary Table 4., Supplemental Digital Content 9). We did include the following sentence in the manuscript to acknowledge that the data used in the analysis is old. “Additionally, the results from this study should be interpreted with caution given the changes in the HIV epidemic and testing tools over the years.”

4. In the introduction, a number of similar previous works are referenced; however, neither there nor more importantly in the discussion are these discussed and the importance of the current work to that body touched on.

Response: We thank the reviewer for this comment. In the introduction we did indicate the gaps in the existing studies stating that most of the studies have relatively smaller sample size, short follow-up time for people with indeterminate results making it impossible to understand the long-term outcomes of people with HIV indeterminate results. We provided justifications for why we think the current study is important for the advancement of HIV testing. 

Throughout the discussion we cited several papers that were initially cited in the introduction section (~8 papers cited in the introduction) to help put our findings in context of the existing literature. For instance, in the discussion we compared the prevalence, correlates and subsequent outcomes of indeterminate results in our study to other studies (citing studies from our introduction). We will be citing additional studies from the studies cited in the introduction to address comment number 5 below.

5. There were several interesting findings, many of which were identified previously, such as males and those married were at higher risk of indeterminate rates. However, this should be discussed further. Why might these issues be happening in those populations?

Response: We thank the reviewer for drawing our attention to this important point. We have added the following sentences to further elaborate on our findings:

“Married women in Rakai are less likely to use contraceptives and more likely to have children(3,4) and parity has been found to be associated with having indeterminate results among females(5). False HIV EIA positive results has also been reported to be associated with multiparous women(6).”

“In Uganda, males have higher prevalence of smoking compared to females(7) and it is well established that cigarette smoking can alter blood viscosity(8). Viscous blood samples or samples with precipitates can form residues which could interfere with ELISA assays(9). This, in part, may explain why males have higher prevalence of indeterminate results compared to females and it is important for public health screening programs to consider this when screening males for HIV infection.”

6. It would also be helpful to understand if any combinations of the factors may result in more indeterminate results. For example, older married males or need those factors always be disaggregated?

Response: We thank the reviewer for the idea of exploring if there is interactive effect. The original goal of this study was not to examine the combined effect of any of the factors. We believe the exploration of the combined effect of multiple indicators and their effect on indeterminate results is a completely different research question which is not a part of the goal of our analysis. This will be considered in future studies.

7. I’m not sure lines 219-238 bring a lot to the results given all others. I might suggest removing, particularly to include a possible analysis per point #6.

Response: Lines 219-238 referenced by the reviewer presents the results for one of the key objectives of the study i.e. the long-term outcomes of people with initially indeterminate results. Removing these lines in our view will defeat the primary goal of the study. 

8. It would also be helpful to more clearly discuss possible reasons for why indeterminate results may be persistent. What could be causing this? Lines 272-279. Also, what can or should be done for those patients?

Response: We added some more some more explanations that possibly result in persistent indeterminate results. We added the following sentence to further explain why indeterminate results may be persistent:

“Some of the factors (systematic lupus erythematosus, rheumatoid factor and polyclonal gammopathy, antibodies to DR-HLA, cross reactivity to core proteins of other retroviruses, mycobacterium leprae infection, in vitro hemolysis, parity and tetanus vaccination(10)) known to be associated with indeterminate results are chronic or permanent which in part may explain why certain people may remain indeterminate for a long period of time.”

9. It also might be worth reflecting on the large sample size. No sample size calculation was included and though some of the differences are relatively small, the incredibly large sample size would cause significance.

Response: We thank the reviewer for this comment. The following sentence has been added to the limitations section: “The large sample could also result in statistical significance.”

Minor comments

1. Line 36: ‘serological’ results rather than ‘serologic’. Also, ‘immunoassays’ is likely the best terminology.

Response: We thank the reviewer for thoroughly reading our paper. The above error has been fixed.

2. The suggestion that those with primary education are less likely to have indeterminate results should be more carefully considered given the adjPR crosses 1.

Response: We thank the reviewer for drawing our attention to this point. We were interpreting the point estimate and we were not necessarily referring to statistical significance in this situation. 

3. Line 58, ‘assay’ should likely be ‘assays’ for both.

Response: We thank the reviewer for thoroughly reading our paper. The above error has been fixed.

4. Lines 189-206 should be double-spaced.

Response: We thank the reviewer for thoroughly reading our paper. The above error has been fixed.

5. Line 259-260: I’m not sure that’s still the case given significant scale-up of HIV viral load testing and tuberculosis testing, both molecular PCR-based.

Response: We thank the reviewer for their thoughts on this. It is true that there are some advances in scaling up PCR-based HIV viral load and tuberculosis testing, but this is still limited to large central hospitals or labs at the national level in several low- and middle-income countries. We do not believe PCR-based testing is available particularly at the subnational level in several LMICs.

---

## [Editor Report · Decision Letter 2]

31 Jul 2020

HIV serologically indeterminate individuals: future HIV status and risk factors

PONE-D-19-22579R2

Dear Dr. Laeyendecker,

We’re pleased to inform you that your manuscript has been judged scientifically suitable for publication and will be formally accepted for publication once it meets all outstanding technical requirements.

Kind regards,

Zixin Wang, PhD.

Academic Editor

PLOS ONE

---

## [Editor Report · Acceptance letter]

14 Aug 2020

PONE-D-19-22579R2 

HIV serologically indeterminate individuals: future HIV status and risk factors 

Dear Dr. Laeyendecker:

I'm pleased to inform you that your manuscript has been deemed suitable for publication in PLOS ONE. Congratulations! Your manuscript is now with our production department. 

Kind regards, 

on behalf of

Professor Zixin Wang 

Academic Editor

PLOS ONE